# Human Herpes Virus-8 Oral Shedding Heterogeneity Is Due to Varying Rates of Reactivation from Latency and Immune Containment

**DOI:** 10.3390/v17111500

**Published:** 2025-11-13

**Authors:** David A. Swan, Elizabeth M. Krantz, Catherine M. Byrne, Fred Okuku, Janet Nankoma, Innocent Mutyaba, Warren Phipps, Joshua T. Schiffer

**Affiliations:** 1Vaccine and Infectious Diseases Division, Fred Hutchinson Cancer Center, 1100 Eastlake Ave. E, Seattle, WA 98109, USA; 2Uganda Cancer Institute, Kampala P.O. Box 3935, Uganda; 3Division of Allergy and Infectious Diseases, Department of Medicine, University of Washington, Seattle, WA 98195, USA

**Keywords:** human herpes virus 8, Kaposi sarcoma, HIV, mathematical modeling

## Abstract

Human herpesvirus-8 (HHV-8) is a gamma herpesvirus linked to the development of Kaposi sarcoma (KS). KS is more common in persons living with HIV (PLWH), but endemic KS in HIV-negative individuals is also common in sub-Saharan Africa. HHV-8 shedding occurs in the oral mucosa and is likely responsible for transmission. The mechanistic drivers of different HHV-8 shedding patterns in infected individuals are unknown. We applied stochastic mathematical models to a longitudinal study of HHV-8 oral shedding in 295 individuals in Uganda who were monitored daily with oral swabs. Participants were divided into four groups based on whether they were HIV-negative or -positive, as well as KS-negative or -positive. In all groups, we observed a wide variance of shedding patterns, including no shedding, brief episodic low viral load shedding, prolonged episodic medium viral load shedding, and persistent high viral load shedding. Our model closely replicates patterns in individual data and attributes higher shedding rates to increased rates of viral reactivation and lower median viral load values to more rapid and effective engagement of cytolytic immune responses. Our model provides a framework for understanding different shedding patterns observed in individuals with HHV-8 infection.

## 1. Introduction

Human herpesvirus-8 (HHV-8) is a gamma herpesvirus that was discovered in 1994 as the causative agent of Kaposi sarcoma (KS) [1], a hyperproliferative endothelium cancer most often observed in individuals with HIV [2]. Endemic KS occurs in HHV-8-positive, HIV-negative individuals and is common in sub-Saharan Africa [3,4]. Yet, most individuals with HHV-8, even those who shed the virus orally frequently and at high viral loads, do not develop KS [5,6,7,8]. The link between oral shedding and transmission, as well as KS tumorigenesis, remains unclear [5,9,10,11].

HHV-8 and Epstein–Barr virus (EBV), the other human gamma herpesvirus, are similar in many aspects. Both establish latency within B cells, while HHV-8 may become latent in epithelial and endothelial cells, allowing the virus to persist for the lifetime of the human host in a non-replicative state [12,13]. HHV-8 and EBV have been implicated in oncogenic disorders of B cells, including multicentric Castleman’s disease (MCD) [14,15,16,17] and primary effusion lymphoma (PEL) [18,19] for HHV-8, and several B cell lymphomas for EBV [20]. HHV-8 and EBV establish lytic infection in both endothelial and epithelial cells [21]. Viral replication in oral epithelial cells leads to substantial amplification of viral DNA and may facilitate viral transmission [5,22].

Viral shedding kinetics provide a key window into host–virus interactions and differ significantly between different human viral pathogens, as well as the immune status of infected individuals. Mathematical models are a critical tool for the analysis of longitudinal viral load data and account for the non-linearity inherent to this data [23]. Models have been applied to capture the timing and intensity of immune responses in tissue [24], to compare different severities of disease among multiple infected individuals [25], and to optimize therapeutic and vaccination approaches [26]. For human herpes viruses, including herpes simplex virus-2 (HSV-2) [27,28], cytomegalovirus (CMV) [29], and EBV [13,30], stochastic models are required, given the episodic and unpredictable nature of shedding within chronically infected individuals due to the latent/lytic cycle of infection. While these models cannot be used to precisely recapitulate an individual’s shedding trajectories, they can capture patterns of shedding using summary statistics such as median and variability of viral load, episode rate, expansion and clearance slope, and peak viral loads [31]. While shedding kinetics differ markedly among human herpes viruses, a commonality is that critical immune responses occur in micro-environments and often over narrow time intervals [5,30,32,33]. Here, we develop the first mathematical models to describe HHV-8 shedding data in the oral cavity to understand observed shedding patterns in a Ugandan cohort of individuals with and without HIV infection and with and without KS [11].

## 2. Materials and Methods

### 2.1. Mathematical Model of HHV-8 Oral Shedding

We developed a stochastic mathematical model to understand the basis of observed heterogeneous HHV-8 shedding patterns in the oral mucosa. Our model focuses on key parameters observed in our recent EBV model, including the rate of viral reactivation from latency and the effectiveness of the peripheral immune responses in eliminating infected cells [30]. Our model also assumes multiple micro-regions to simulate the oral cavity based on analysis performed in previous modeling of EBV [30]. Each region is assumed to have a certain probability of viral reactivation and a separate immunologic response (Figure 1). Minor additions to the original EBV model are described below and include the addition of an eclipse phase during which infected cells are not yet producing a virus, as well as differentiation between a cell-associated virus and a cell-free virus, only the latter of which is detected with swabs.

To limit model parameters, we assigned an infectivity to each infected cell (I), assuming a certain rate of viral production, rather than modeling the infectivity of each virion. The model includes a reactivation rate *ϕ*, which dictates how frequently latently infected cells become lytic. When lytic infection is initiated, an infected cell begins producing viruses and infecting adjacent cells. While the reactivating cell may be a B cell, B cells and reseeding of the latently infected cell reservoir are not explicitly included as model variables due to the lack of available data to parameterize the dynamics of these cell types separately. After an eclipse period (E) of average duration 1/*ε*, the infected cell begins producing a cell-associated virus (V_c_) at a rate of *p* virions per day. The infection spreads to additional cells in the region based on the infectivity parameter *β*.

As the population of infected cells in a region increases, the population of local HHV-8 specific T cells (T) may increase and kill infected cells at rate *f*. Infected cells also die at rate *a*. When the cell dies, the cell-associated virus is released from the cell and becomes a cell-free virus (V_e_). V_e_ is captured during daily swabbing of the oral cavity. The cell-free virus then decays at rate *c*. The T cell population in each region grows at a maximal rate of *θ*. The rate is modulated by the number of infected cells in the regions, with T cell growth being half maximal at a level of *r50* infected cells. The *r50* parameter represents a threshold of infected cells sufficiently high for antigen presentation that stimulates a T cell response. T cells are also born within each region at rate α and die at a rate of δ.

The initial T cell population in each region is randomly seeded and then allowed to equilibrate by running the model for a year before fitting to data. This step allows for a variable set of immune starting conditions across micro-regions, which is, in turn, dependent on input parameter values. Biologically, this implies that for a given set of virologic and immunologic parameters, the model will establish spatially heterogeneous immune conditioning during the transition from an acute to chronic, established infection. For the purpose of data fitting, we are typically modeling established infection in our system. Of note, after a year of equilibration, the model output, as shown in Figure 2, is stable for a given parameter set when the model is simulated repeatedly for 30-day intervals, proving that the model equilibrates after a year.

The stochastic equations for the model are shown below. Each region is represented using the subscript j. The major compartments are E_j_ for infected cells in the eclipse phase, I_j_ for productively infected cells, T_j_ for T cells, V_cj_ for cell-associated viruses, and V_ej_ for extracellular viruses. V_etot_ is the amount of the virus available for oral swabbing at each time step and is the sum of all regions. System reactions within a single micro-environment *j* are as follows:E_j_ ⟶ E_j_ + 1 with rate (*β* × I_j_) + *ϕ*;E_j_ ⟶ E_j_ − 1 with rate *ε* × E_j_;I_j_ ⟶ I_j_ + 1 with rate *ε* × E_j_;I_j_ ⟶ I_j_ − 1 with rate (*a* × I_j_) − (*f* × I_j_ × T_j_);T_j_ ⟶ T_j_ + 1 with rate *α* + (F(I_j_) × θ × T_j_);T_j_ ⟶ T_j_ − 1 with rate *δ* × E_j_;F(I_j_) = I_j_/(I_j_ + r50);V_cj_ ⟶ V_cj_ + 1 with rate *p* × I_j_;V_cj_ ⟶ V_cj_ − 1 with rate (*a* + (*f* × T_j_)) × V_cj_;V_ej_ ⟶ V_ej_ + 1 with rate (*a* + (*f* × T_j_)) × V_cj_;V_ej_ ⟶ V_ej_ − 1 with rate *c* × V_ej_;V_etot_ = V_e1_ + V_e2_ + … +V_e240_.

The model does not include viral spread between the 240 modeled micro-environments. When multiple regions are involved during an episode, it is, therefore, due to multiple stochastic sites of reactivation from latency. We selected 240 regions based on prior EBV models of the oral epithelia [30].

We also tested a model where virions initiate the spread of infection among cells in each region and where cells in one region can infect cells in adjacent regions. The latter did not fit the data as well despite its added complexity.

### 2.2. Cohort Data

We fit the model to data from an observational, prospective cohort study of Ugandan adults (age ≥ 18 years) who were enrolled between October 2007 and May 2010. Individuals were from four groups: (A) HIV-1 seropositive individuals without KS (HIV+/KS-); (B) HIV-1 seronegative individuals without KS (HIV-/KS-); (C) HIV-1 seropositive individuals with KS (HIV+/KS+); and (D) HIV-1 seronegative individuals with KS (HIV-/KS+). HIV+/KS- participants who reported ART use at the time of enrollment were excluded. HIV+/KS+ participants were eligible regardless of ART use. At the time of the original study, ART was initiated based on CD4+ T cell levels, and universal test and treat policies were not yet established. Participants from all arms were followed for at least 4 weeks, with one “session” of data collection consisting of 28 days of daily home oral swab collection. This monthly cycle of sample collection was repeated every three months for Arm A (HIV+/KS-) participants (up to two years) and for Arm C (HIV+/KS+) participants (up to one year) [34].

Oral swab samples were evaluated for HHV-8 DNA by a quantitative, high-throughput fluorescent-probe-based real-time polymerase chain reaction (PCR) assay (TaqMan assay, Applied Biosystems, Foster City, CA, USA) of the ORF73 gene at the UCI-Fred Hutch Cancer Centre Laboratory in Kampala, Uganda, as described previously [9,10]. Samples with >100 copies per swab of HHV-8 DNA were considered positive [35].

### 2.3. Model Fitting

Our model simulations are intended to match data from 295 individuals enrolled in the natural history cohort study who had at least one positive oral sample during the initial 4-week sampling period. Rather than fit to individual viral trajectories, we attempted to reproduce three summary measures from each person: shedding rate (percentage of swabs positive for HHV-8 DNA by PCR), median log HHV-8 viral load per positive swab, and peak log HHV-8 viral load. To match the study protocol, the simulation data was sampled daily for 28 consecutive simulated days. The model was simulated using a tau-leap algorithm with a time step of 0.05 days (72 min).

Parameters for the models were fixed when possible, using values from the literature or fitted to match the observed data. All fixed and fitted parameters are shown in Table 1, together with their values and references.

Initially, fitted parameters were drawn from wide, primarily log-linear distributions. One thousand sets of simulations were run, each with a different set of values for the fitted parameters. Adaptive immunity levels were established by running the simulation for one year and measuring T cell levels in each region. Since the simulations were run stochastically, 10 runs were performed for each unique set of parameters. Ten runs were sufficient to generate consistent output metrics for data comparison, as shown in Figure 2. To limit computational run times and because we achieved convergence of model output, we limited the selection process to two sequential rounds of fitting. The same starting T cell distributions were used in each run to save computational time, but sampling was delayed until day 150 to allow for ongoing episodes to be captured. The median and peak log HHV-8 viral loads and shedding rates were averaged across runs. The shedding rate was defined as the total number of samples with detectable viral DNA (>100 genomic copies per swab) divided by the total number of samples.

Each of the 1000 simulations was scored against the data for that individual. The score was composed of the average of the percent error (the absolute difference between simulated and target values as a percentage of the target value) for each of the three measures. The parameter values for the top 100 runs were used to determine a posterior distribution for each parameter, which was defined using their values for mean and standard deviation. Those distributions were used to draw 1000 new parameter combinations, and the process was repeated. After two such rounds of simulations to narrow distributions, the mean values of each fitted parameter were recorded. These were then used in a set of follow-on simulations (10 per participant), with the average simulated metrics used to determine concordance between simulated and actual data across all participants. We used the longest observed episode duration and episode rate as post hoc shedding features for further model testing, independent of the fitting process. All code is available here: https://github.com/FredHutch/HHV8_Inf_Cell_Model.git. The URL was accessed on 28 May 2025.

## 3. Results

### 3.1. Cohort Summary Statistics

Thirty-six (29%) HIV+/KS- participants, sixteen (21%) HIV-/KS-, fifty-six (74%) HIV+/KS+, and nine (50%) HIV-/KS+ participants had HHV-8 detected in at least one oral swab. Data from these individuals was used for model fitting. As we described in an earlier analysis of oral HHV-8 shedding in the cohort study, shedding rates and median viral load varied substantially across individuals in all four groups and were correlated with one another [34].

### 3.2. Model Simulations Recapitulate Observed Variability in HHV-8 Shedding

Mathematical model output showed high correlation and concordance with the observed shedding patterns across all individuals with positive swabs during the study period, as shown in Figure 2. The concordance correlation coefficient (CCC), a measure of correlation and agreement between simulated and observed data, was 0.988 for the shedding rate (Figure 2A), 0.915 for median log HHV-8 (Figure 2B), and 0.922 for peak viral load (Figure 2C). Modeled peak viral load slightly exceeded observed low-peak episodes and was often slightly lower than the high observed high peaks (Figure 2C). Pearson correlation coefficients were also high for each metric (Figure 2A–C). Each parameter combination was also scored using the average percentage error between each individual’s shedding metrics and simulated output as described in the Methods. Averaging the scores for the top 10 parameter combinations per participant across the three categories, the average percent error was 3.7%, indicating that the model output was very close to the cohort data.

We further validated the model by assessing for correlation between two other metrics that were not used to fit the model. The episode rate, with episodes counted as ongoing if present at sampling onset or new if followed by a negative sample, had a Pearson correlation coefficient of 0.7 (*p* < 0.001). Longest episode duration, defined as the longest duration of positive samples during the interval, had a Pearson correlation coefficient of 0.92 (*p* < 0.001). These coefficient values were likely lower than those observed in Figure 2 because these metrics reflect stochasticity in the data and model output.

We previously demonstrated in the cohort study that HHV-8 shedding occurs along a gradient [34]. On one extreme, certain individuals shed at a rate exceeding 80%, typically at high median viral loads > 10^5^ copies/swab, which may vary over time over a range of several logs. Other infected people shed episodically at a low rate (<20%), typically at much lower median viral loads (<10^3^ copies/swab), and rarely with a single sample exceeding 10^5^ copies/swab. Many individuals have shedding patterns falling between those two extremes.

HHV-8 viral load traces were plotted from different representative participants in each of three different shedding categories: infrequent shedding (one or two positive swabs during the 28 days; Figure 3A and Appendix A), medium shedding (~30–70% of the swabs being positive; Figure 3C and Appendix A), and constant shedding (all swabs positive; Figure 3E and Appendix A). The simulation run plots show patterns compatible with observed shedding (Figure 3B,D,F and Appendix A) despite the stochastic nature of the model output. Individuals with high shedding rates also tended to have higher median viral loads (Figure 3G), a trend that was also observed in the simulated output (Figure 3H).

### 3.3. Mechanistic Predictions of HHV-8 Shedding Rate and Viral Load

We next used our simulated output to determine how mathematical model parameter variability correlated with key shedding outcomes. We observed that a higher viral reactivation rate was highly correlated with a participant’s HHV-8 shedding rate, while the per-cell viral production rate was also somewhat positively correlated (Figure 4A,C,E). A higher viral per-cell production rate was most predictive of higher median viral load, though the viral reactivation rate was somewhat positively correlated and the immune killing rate was somewhat negatively correlated (Figure 4B,D,F). Overall, all three model inputs had at least a moderate impact on the shedding rate and median viral load.

### 3.4. Parameter Ranges According to HIV and KS Status

We observed high variability among all parameter values for all study groups (Figure 5). The average immune cell killing rate was slightly higher for the HIV-/KS+ groups than for other groups (Figure 5A). The mean per cell viral production rate was the highest in HIV-/KS– individuals and the lowest in HIV-/KS+ individuals (Figure 5B). We also observed slightly higher HHV-8 reactivation rates in the HIV-/KS+ group versus the other groups (Figure 5C).

## 4. Discussion

We used a mathematical model to capture HHV-8 shedding kinetics in a diverse cohort of individuals stratified by HIV-1 infection and KS status. The model’s stochastic output does not allow precise recapitulation of individual oral shedding patterns but does reproduce key metrics of individuals in the cohort, including very close approximations of the shedding rate, median viral load, and peak viral load. The simulated data is, therefore, an accurate representation of the true data at the individual level and across the cohort.

Each human herpesvirus is notable for a specific pattern of shedding, which differs according to numerous factors, including the viral detection rate, episodicity versus persistence, viral expansion rate, viral clearance rate, median viral load, peak viral load, and stability over various time intervals. We have applied mathematical models to most of these viruses to develop hypotheses regarding their observed shedding kinetic profile [27,28,29,32,38]. We consistently identified that herpes virus shedding is ultimately driven by rates of viral reactivation from latency, rates of viral amplification in the lytic epithelial cell compartment, and the pace of the antiviral immune response. A shared feature among viruses, including HSV-1 and -2 [27,28], human herpes virus-6 (HHV-6) [32,38], CMV [29,39], and EBV [30], is that variable observed shedding rates and the duration of shedding events can be attributed to the latency reactivation rate and rate of peripheral immune control. Our current modeling suggests that HHV-8 appears to follow these trends as well.

Many specific features of HHV-8 shedding closely resemble EBV shedding [30,34], while differing significantly from the other human herpesviruses [27,28,29,32,38,39]. Specifically, we observe an extremely wide range of HHV-8 shedding phenotypes, including rare and brief episodic shedding, more frequent and prolonged episodic shedding, and continual high viral load shedding. These distinct patterns are not observed for other human herpesviruses, including HSV-2, CMV, and HHV-6 [30,34]. For both EBV and HHV-8, the shedding rate is correlated with the median viral load [30,34].

Our model explains this heterogeneity according to variance in key biological parameters. As with EBV, the rate of viral reactivation from latency is predicted to be highly associated with the observed shedding rate, while median viral load appears to have multi-factorial determinants, including the viral reactivation rate, the rate of viral production from infected cells (which may relate to innate immunity), and rate of killing by the acquired/memory arm of the immune system. In theory, each of these processes would be a viable therapeutic target, but even a partial reduction in the rate of viral reactivation from latency might have a significant effect on shedding and possible transmission likelihood.

Of note, several alternative mechanisms that could explain variable viral loads and shedding rates are not included in the model. We do not include target cell limitation or variability in cellular susceptibility to infection as a mechanism because epithelial and endothelial cells are present in high abundance in the oral mucosa, active infection does not appear to lead to large lytic lesions, and the latent–lytic cycle is fundamental to herpes virus biology [12,13]. Regarding immunity, we assume cytolytic memory responses compatible with T cell activity but cannot rule out the possibility of innate response variability explaining heterogeneous shedding patterns.

As described in our previous work, there did not appear to be a clear set of shedding or model patterns that clearly differentiate infected individuals with and without HIV-1 and with and without KS [34]. This is because in each of the four groups stratified according to these criteria, we observe all types of shedding [34]. This observation does not preclude the possibility that oral shedding could be a valuable surrogate for the development of KS, as well as treatment response in established KS. More work is required to establish whether any metric of HHV-8 shedding can be used as a surrogate marker for risk of progression to cancer or to monitor treatment response.

The major limitation of our work is that the model is quite simple and does not capture the complexity of HHV-8 latent and lytic infection. We do not ascribe a cellular source to viral reactivation from latency, for viral lytic replication, or for the acquired immune response based on a lack of sufficient longitudinal, quantitative data for model development. There is considerable knowledge about molecular mechanisms dictating the balance between HHV-8 latency and reactivation in an infected cell [40]. Our model suggests that drugs that either sustain latency or eliminate the source of HHV-8 reactivation could be of high therapeutic value, but a major priority is to establish the nature and site of these cells in vivo. Similarly, our model attributes high importance to peripheral immunity in determining viral load but does not discriminate whether antibodies, T cells, or some combination mediate this effect, as this data is also lacking. This will be important to discern to help guide the use of immunotherapy in KS and other HHV-8-related diseases.

In summary, we have created the first model of HHV-8 shedding. It performs well in terms of reproducing detailed shedding kinetics across a broad range of shedders. Future work is needed to inform mechanisms of viral latency and immune control to allow more detailed modeling.

## Figures and Tables

**Figure 1 viruses-17-01500-f001:**
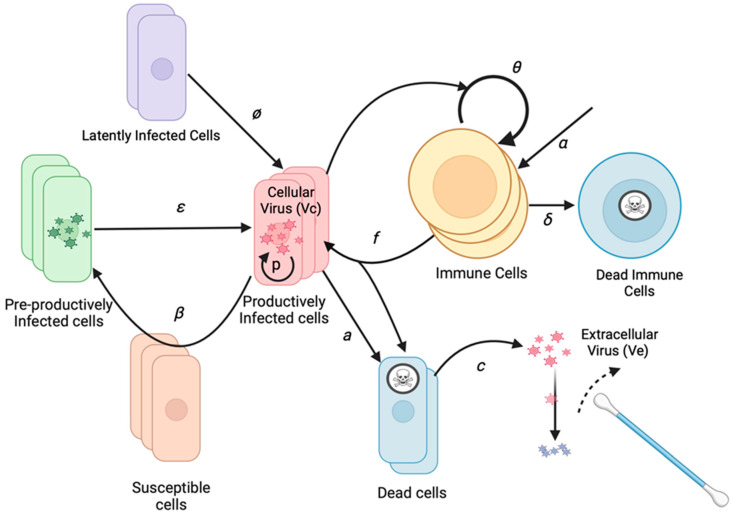
HHV-8 mathematical model schematic of an infection within the oral mucosa. Basic model assumptions are that latently infected cells reactivate at rate *ϕ*. Infection spreads from cell to cell with infectivity *β*. Cells exit an initial non-productive eclipse phase at rate ε. Infected cells die after becoming lytic at rate *a*. Infected cells produce a virus at rate *p*. Cytolytic immune cells expand at rate *θ* and become half maximal when *r50* cells are infected, kill infected cells at rate *f*, which are produced independently of infected cells at rate α, and die at rate δ. The cell-associated virus converts to a cell-free virus upon cell lysis and then decays at rate *c*. The cell-free virus is detected with oral swabs. Multiple regions can produce viruses concurrently.

**Figure 2 viruses-17-01500-f002:**
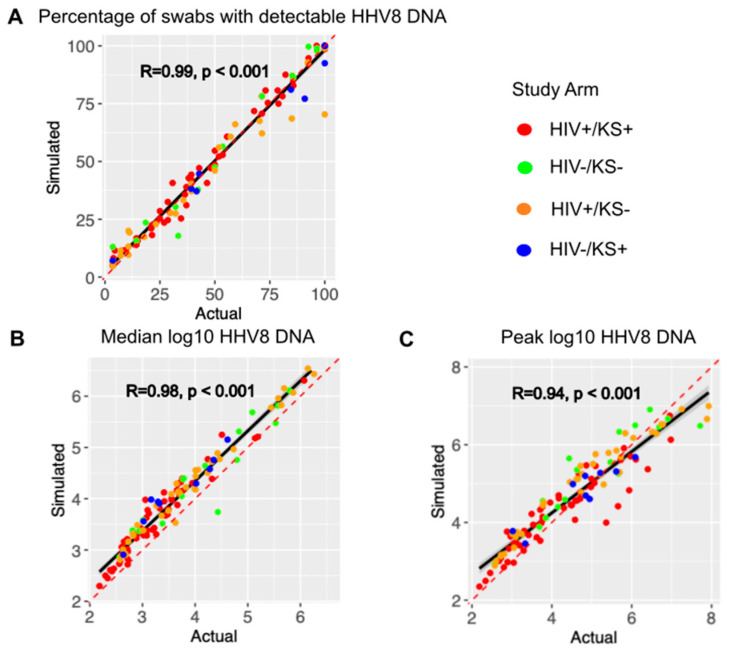
The mathematical model output reproduces individual shedding metrics. Actual (*x*-axis) versus simulated (*y*-axis) for (**A**) the percentage of positive swabs, (**B**) median log10 HHV-8 DNA copies, and (**C**) peak log10 HHV-8 copies. Each dot is a study participant. The red dotted line is the line of perfect concordance. The solid black line is a concordance line of actual versus simulated data. Participants are colored by the study arm in the legend. R is the Pearson correlation coefficient. *P* is the statistical significance.

**Figure 3 viruses-17-01500-f003:**
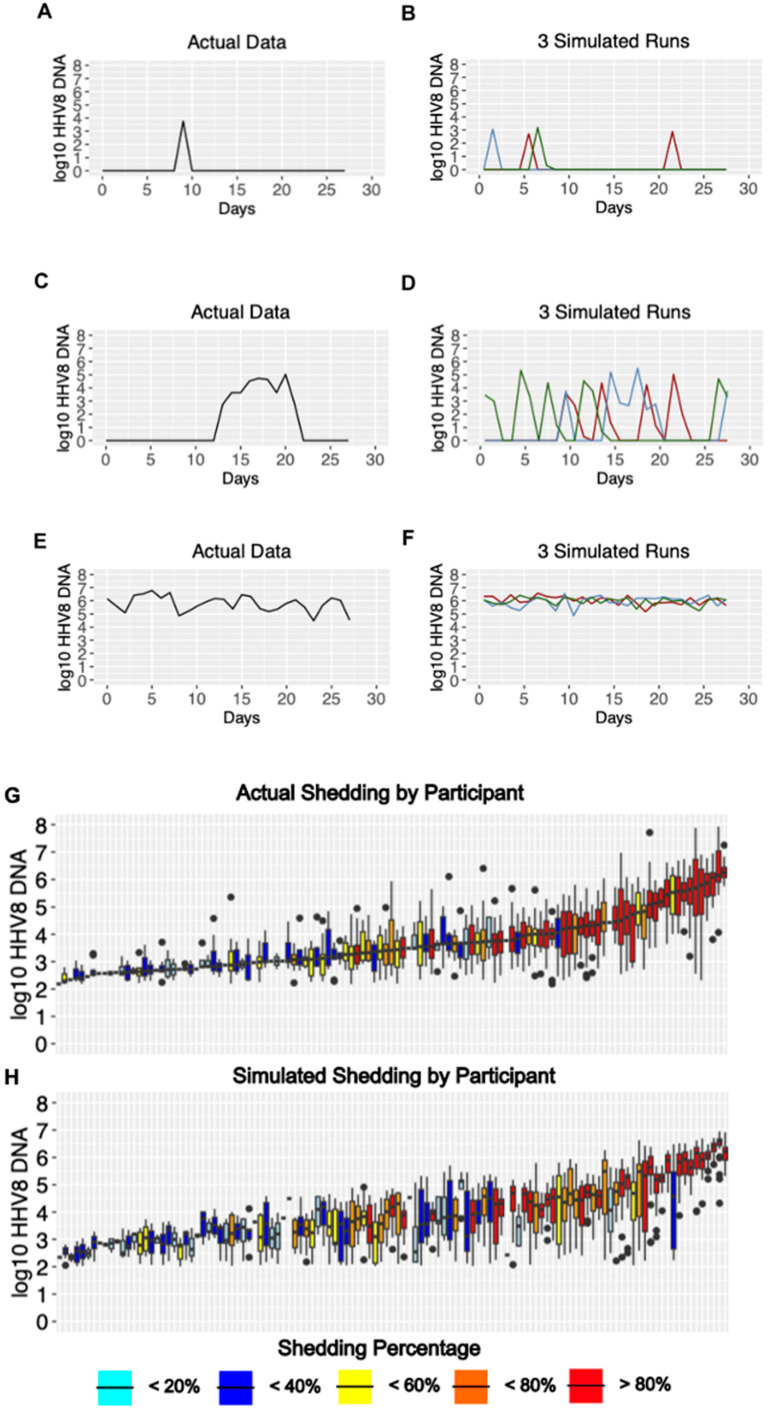
Model output matches three observed HHV-8 shedding types. Study data from three participants showing (**A**) rare episodic shedding, (**C**) more prolonged episodic shedding, and (**E**) constant higher viral load shedding is matched in a general fashion by repeated model simulations in (**B**,**D**,**F**), respectively (three per parameter set indicated with differing color lines). The low threshold cutoff was 100 DNA copies per swab. Individual data and simulations are shown in Appendix A. (**G**,**H**). Distribution of oral HHV-8 viral loads for actual and simulated study participants. (**G**) Actual participant data sorted from left to right by median viral load and (**H**) simulated participant data matched vertically to actual data in (**G**). Each boxplot shows data from a participant; boxes represent the interquartile range, horizontal lines within boxes represent medians, and whiskers extend to 1.5x the interquartile range. Dots fall outside of this range. Participants with only a median bar and no box had too few positive samples to generate an interquartile range. Participants are color-coded according to shedding rate.

**Figure 4 viruses-17-01500-f004:**
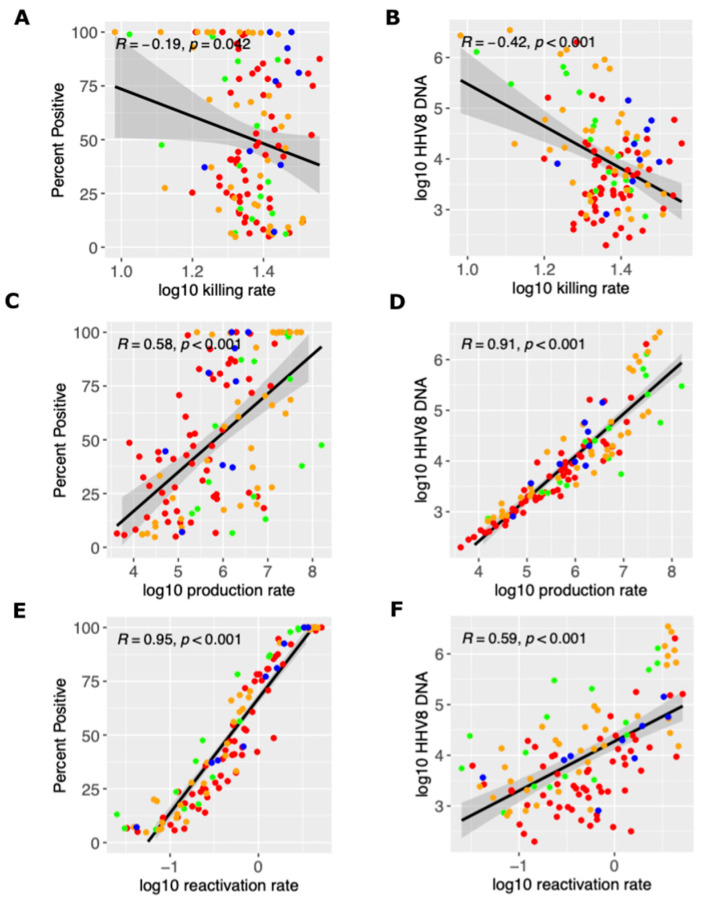
Mathematical model and mechanistic predictors of the shedding rate and median viral load. Possible determinants of outcomes are x-axes and include (**A**,**B**) the log10 killing rate of infected cells, (**C**,**D**) the log10 viral production rate per cell, and (**E**,**F**) the log10 viral reactivation rate from latency. Y-axes are outcomes that include (**A**,**C**,**E**) the shedding rate and (**B**,**D**,**F**) median viral load. Each dot is a simulated participant with red (HIV+/KS+), green (HIV-/KS-), orange (HIV+/KS-) and blue (HIV-/KS+). Black line is correlation line with 95% shaded confidence interval.

**Figure 5 viruses-17-01500-f005:**
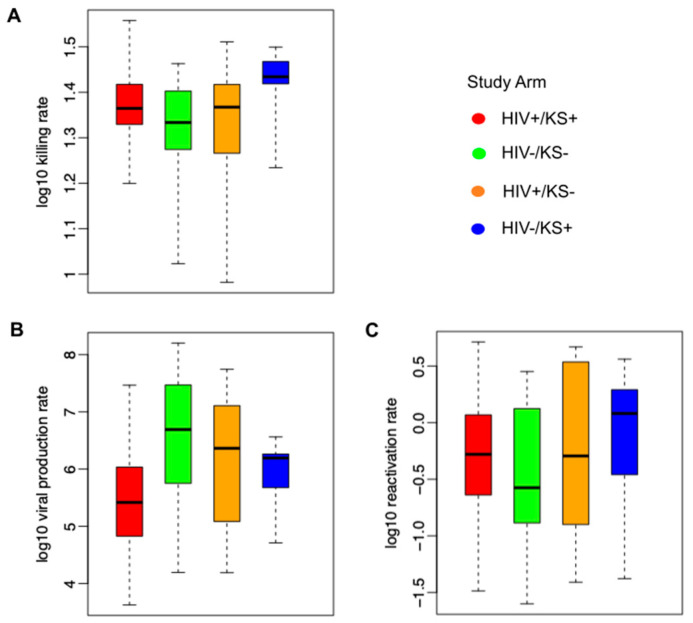
Mathematical model parameter ranges according to HIV and KS status. (**A**) Ranges of model-derived immune cell killing rates, (**B**) per cell viral production rates, and (**C**) latency viral reactivation rates. Each boxplot shows model-estimated parameters (mean values estimated from the posterior distribution) from all participants in each group; boxes represent the interquartile range, horizontal lines within boxes represent medians, and whiskers extend to the minimum and maximum values.

**Table 1 viruses-17-01500-t001:** Model parameters.

Parameter	Symbol	Value	Reference
Max T cell growth rate	*θ*	3/day	[28,36]
Eclipse period	*ε*	0.85 days	[37]
T cell birth rate	*α*	0.312/day	* Fitting (range 0.1–0.5)
T cell death rate	*δ*	0.001/day	[28]
Infected cells for half-maximal T cell growth	*r50*	300 cells	* Fitting (range 2–3.5 log10)
Viral clearance rate	*c*	6/day	* Fitting (range 5–20)
Infected cell death rate	*a*	mean = 1.48/day sd = 0.01	Fitting (range 0.5–2.5)
Cell infectivity	*β*	mean = 8.2 cells/day sd = 1.4	Fitting (range 1–20)
Viral production	log *p*	mean = 5.86 log10 viruses/cell/day sd = 1.1 log10	Fitting (range 3.5–7 log10)
T cell killing rate	*f*	mean = 23.5 cells/day sd = 24	Fitting (range −2–2 log10)
Reactivation rate	*ϕ*	mean = 1.08 cells/day sd = 1.7	Fitting (range −1–2 log10)

* Some of the fixed parameters were initially fitted and later fixed, as their final best ranges proved to be quite narrow. Varying these parameter values within two standard deviations of the mean did not affect fitting metrics shown in Figure 2. Other fixed parameter values were gathered from previous studies. These included the birth rate of T cells, the level of infected cells at which T cell growth is half maximal, the T cell exhaustion rate, and viral clearance rates. Limiting the set of fitted parameters allowed more in-depth exploration of their parameter space.

## Data Availability

All raw data use for modeling can be visualized here https://academic.oup.com/ofid/article/11/10/ofae548/7764581 (accessed date 7 November 2025). and will be made available upon email request to the corresponding author.

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
