# Peer review of "Human Herpes Virus-8 Oral Shedding Heterogeneity Is Due to Varying Rates of Reactivation from Latency and Immune Containment"

_viruses, 2025, doi:10.3390/v17111500_

Round 1
Reviewer 1 Report
Comments and Suggestions for Authors
The paper by David A Swan et al. presents a mathematical model to characterize HHV-8 oral shedding kinetics.
Here are the minor revisions I recommend before publication:
Lines 67-72 (par. 2.1): please add some detail, for example a brief explanation of the model previously published.
Line 75 (caption fig. 1): I suggest to write the symbol in figure and in caption in the same way.
Line 105: Vetot, it will be better if “tot” is subscript (Vetot)
Line 116: “Individuals were from four groups: HIV-1 seropositive individuals without KS…”. Add also the call for arm (A, B, C, D) stated later in the text.
Line 119: “HIV+/KS+ participants were eligible despite ART use.” Please, briefly explain why this distinction was made (why HIV+/KS+ participants were eligible despite ART use)
Line 126 and 130: please substitute KRSV with HHV-8
Line 130: “150 copies per mL” . Is it correct? Or is it per swab?
Paragraph 2.3: please correct HHV8 with HHV-8
Lines 142-143: The authors should clarify which parameters are considered in the observed data and, which are the methods used for collecting the observational data.
Line 218: 150 DNA copies per sample: do the authors mean per swab?
Line 310: the link does not work. From the reviewer’s page (on-line) only figures S1,2,3 are available. Are there also tables or videos?
Author Response
The paper by David A Swan et al. presents a mathematical model to characterize HHV-8 oral shedding kinetics.
Here are the minor revisions I recommend before publication:
Lines 67-72 (par. 2.1): please add some detail, for example a brief explanation of the model previously published.
Thank you. We have added detail to address the structure of our past models, particularly of EBV.
Line 75 (caption fig. 1): I suggest to write the symbol in figure and in caption in the same way.
Thank you. We have changed this accordingly.
Line 105: Vetot, it will be better if “tot” is subscript (Vetot)
We have changed this accordingly.
Line 116: “Individuals were from four groups: HIV-1 seropositive individuals without KS…”. Add also the call for arm (A, B, C, D) stated later in the text.
Good idea. We have made this change,
Line 119: “HIV+/KS+ participants were eligible despite ART use.” Please, briefly explain why this distinction was made (why HIV+/KS+ participants were eligible despite ART use)
This decision was made based on the fact we needed to include individuals on ART to maximize enrollment into this arm (because ART was the standard of care at the time of the cohort). This is now clarified.
Line 126 and 130: please substitute KRSV with HHV-8
Noted and done.
Line 130: “150 copies per mL” . Is it correct? Or is it per swab?
Both are correct because the swab is dipped into 1 mL of fluid. However, we prefer per swab.
Paragraph 2.3: please correct HHV8 with HHV-8
Done.
Lines 142-143: The authors should clarify which parameters are considered in the observed data and, which are the methods used for collecting the observational data.
Thank you. Unfortunately, heterogeneity of T cell density is unknown in infected tissue, and we now mention this.
Line 218: 150 DNA copies per sample: do the authors mean per swab?
Yes. Thank you
Line 310: the link does not work. From the reviewer’s page (on-line) only figures S1,2,3 are available. Are there also tables or videos?
We will ask the editors to fix the link so our supplementary figures which are quite important are available to the reader.
Reviewer 2 Report
Comments and Suggestions for Authors
Swan et al. presented a stochastic differential equation model to study HHV8 oral shedding. The manuscript is generally well-written with interesting methodology.
Major comments:
- The authors made a point to compare HHV8 with EBV throughout the manuscript; however, no quantitative information is provided for EBV and the comparison is completely qualitative. Specifically, they did not provide which results (table, fig) support their comparisons.
- I feel that the conclusion that the "heterogeneity is due to varying rates of reactivation from latency and immune containment" is not well supported. This is because the model does not consider other factors such as susceptible cell dynamics, different immune mechanism, etc. Thus, it is unclear whether in a model that encompasses these other factors, their result would hold. Furthermore, the phrase "immune containment" is not very specific, because they only consider cytolytic effect of T-cells.
- L138, there seems to be an equation used to convert VL to swabs positivity not included in the current manuscript. If so, please provide it with supporting references. This is quite important as it pertains to the very high correlation results in Fig 2A.
- Regarding the use of only median VL per positive swabs (only), and peak VL, this could create a scenario where both of these results match well (Fig 2B-C) without the model actually capturing VL patterns (as observed in Fig 3A-D and SM figures). Have you considered also constraining model fit by using the number of reactivation events, number of peaks, or duration of sustained VL?
- How is this model simulated? For example, do you use the tau-leaping algorithm, etc.?
- Your fitting method feels roughly like a two-step approximate Bayesian algorithm (L157-167), but not quite since you don’t have a threshold to assess for convergence (only used the top 100 runs). Have you examined it convergence? Was this method introduced elsewhere?
- Could you add statistical comparison to 3.4? Also, since you get a “posterior distribution” of the fitted parameters, how is that being converted to the bar plot in Fig 3?
Minor comments:
- L94, please explain infected cells "cease viral production" without dying
- L94, please clarify "At that same rate"
- L98, please edit "with it being half maximal at a level of r50 cells" and also define "r50"
- L100, I don't understand what you mean by "equilibrate by running the model for a year". What do you mean to "equilibrate"? Why do we need that?
- Why 240 regions? That seems arbitrary.
- What is the function K(Ej)? It is not used anywhere in the model.
- The stochastic differential equation model is written in a non-standard way, where the “differential equation” part is written as “difference”.
- L126, KSHV is not defined.
- Table 1 has alpha as the birth rate, but it is defined as the “arrival rate” on L98.
- L146, could you elaborate on “were initially fitted and later fixed”? How insensitive must a parameter be to be fixed after fitting? What about a sensitivity test?
- L154, why 10 runs? This seems like a very small number especially for a relatively more complex model with more than a few processes.
- L154, is this what you mean “The same starting T cell distribution was used …”?
- L251, should “data” be “model estimated parameter”?
- My opinion is that phrases like “extremely closely” (L258), “close representation” (L259), “extremely well” (L305) are not supported by the current results.
- L273-274 needs supporting reference.
- L283, what is “outsized”?
- L285-287, did you do any quantitative or qualitative test to check for similarity in patterns, such as correlation with temporal shift, dynamic time warping, etc.?
Author Response
Swan et al. presented a stochastic differential equation model to study HHV8 oral shedding. The manuscript is generally well-written with interesting methodology.
We appreciate the Reviewer’s generally positive feedback.
Major comments:
- The authors made a point to compare HHV8 with EBV throughout the manuscript; however, no quantitative information is provided for EBV and the comparison is completely qualitative. Specifically, they did not provide which results (table, fig) support their comparisons.
Thank you for this important point. In the revised version, we emphasize that HHV8 shedding resembles EBV shedding in the following ways. First, both viruses have 3 general shedding patterns which include rare episodic shedding, more frequent episodic shedding consisting of longer episodes, and persistent viral shedding at higher viral loads. Second, for both viruses, the shedding rate is correlated with higher median final viral load. Third, an nearly equivalent structured and parameterized mathematical model does a nice job of reproducing viral kinetic patterns for both viruses. Of note, these are not characteristics of other human herpesviruses kinetics (including HSV-1, HSV-2 and CMV).
- I feel that the conclusion that the "heterogeneity is due to varying rates of reactivation from latency and immune containment" is not well supported. This is because the model does not consider other factors such as susceptible cell dynamics, different immune mechanism, etc. Thus, it is unclear whether in a model that encompasses these other factors, their result would hold. Furthermore, the phrase "immune containment" is not very specific, because they only consider cytolytic effect of T-cells.
This is a fair point, and we have added language to the discussion to emphasize that variability in susceptible cell dynamics may be an alternative explanation for observed shedding patterns. We selected our model for two simple reasons. First, a defining and fundamental property of human herpesviruses is periodic reactivation from latency. It would therefore be problematic to leave this mechanism out of the model. Second, HHV8 is able to replicate in a wide range of predominant host cells in the oral mucosa including epithelial cells and endothelial cells. Therefore, we tend to doubt that target cell limitations or variable susceptibility is a major factor in determining heterogeneity between individuals. Note also, that we remain agnostic about the immune mechanisms driving the observed data, other than to say that local immune memory appears to be of high importance.
- L138, there seems to be an equation used to convert VL to swabs positivity not included in the current manuscript. If so, please provide it with supporting references. This is quite important as it pertains to the very high correlation results in Fig 2A.
We apologize for not defining shedding rate which is simple the percentage of positive swab or number of positive swabs with HHV8 > 100 DNA copies divided by the total number swabs. Herpes virus shedding rate is the most common metric used in clinical trials of antiviral drugs.
- Regarding the use of only median VL per positive swabs (only), and peak VL, this could create a scenario where both of these results match well (Fig 2B-C) without the model actually capturing VL patterns (as observed in Fig 3A-D and SM figures). Have you considered also constraining model fit by using the number of reactivation events, number of peaks, or duration of sustained VL?
This is an excellent point. We analyzed the Pearson correlation for episode rate (number of reactivation events) and duration of longest observed episode. These remained high (0.70 and 0.92 respectively). We suspect these values are lower than the three metrics included in Figure 2 because the metrics pertain to individual episode characteristics rather than the entirety of the observed data. Nevertheless, these results which are included in the text of the revised paper further validate the model. We apologize for not including number of peaks in the analysis as we no longer have a staff member working on this project and this would involve a significant analysis beyond the scope of our capabilities at present.
- How is this model simulated? For example, do you use the tau-leaping algorithm, etc.?
Thank you for the question. The model is simulated using a tau-leap algorithm with a time step of 0.05 days or 80 minutes.
- Your fitting method feels roughly like a two-step approximate Bayesian algorithm (L157-167), but not quite since you don’t have a threshold to assess for convergence (only used the top 100 runs). Have you examined it convergence? Was this method introduced elsewhere?
The decision to run two rounds selecting the top 10% of runs to redefine the parameter ranges was made primarily based on compute time and convergence after these two steps. Long run times were driven by the larger number of simulated participants (117) and the number of competing models. We noted strong trends of convergence among the top 10% of tuns towards better model fits for all model metrics in Figure 2.
- Could you add statistical comparison to 3.4? Also, since you get a “posterior distribution” of the fitted parameters, how is that being converted to the bar plot in Fig 3?
Figure 3 does not include any parameter information. We think the reviewer is referring to Figure 5. In Figure 5, we prefer not to show statistical comparisons these results are hypothesis generating rather than testing. The bar plots are generated using the mean of the posterior distribution.
Minor comments:
- L94, please explain infected cells "cease viral production" without dying.
We cannot tell whether the cells actually die or merely stop producing virus due to the immune response. This is simplified in the revision.
- L94, please clarify "At that same rate".
We explain that the viral release from cells is assumed to occur when cells lyse. Therefore, the rate is the same.
- L98, please edit "with it being half maximal at a level of r50 cells" and also define "r50"
Thank you. We have clarified this section.
- L100, I don't understand what you mean by "equilibrate by running the model for a year". What do you mean to "equilibrate"? Why do we need that?
Thank you. We have now added a full paragraph to explain this. A key feature of our group’s herpes models is that they are very sensitive to initial immune conditions. For a given set of parameters, we run the model for a year which allows these conditions to equilibrate. Model output (such as shedding rate, median viral load and peak viral load assessed in 30-day intervals is highly auto-correlated and consistent from that point forward given no change in assumed parameter values. We believe that this one year run in roughly corresponds to acute infection converting to chronic persistent infection, during which a détente is established between the virus and host immune response. This has been well established for HSV-2 infection, but we conceded requires more research for HHV8.
- Why 240 regions? That seems arbitrary.
We acknowledge that this is somewhat arbitrary and was based on past models of EBV infection. For similarly structured models of HSV infection, we have shown that equivalently good model fits are possible if we assume 2 times less or more regions. We did not have time to perform these analyses for this work as the first author has retired.
- What is the function K(Ej)? It is not used anywhere in the model.
This function should not be included and has been removed. Thank you for noticing this.
- The stochastic differential equation model is written in a non-standard way, where the “differential equation” part is written as “difference”.
Thank you. We agree and have changed the notation.
- L126, KSHV is not defined.
Thank you. We now change this to HHV-8.
- Table 1 has alpha as the birth rate, but it is defined as the “arrival rate” on L98.
Thank you. We now change it to birth rate throughout.
- L146, could you elaborate on “were initially fitted and later fixed”? How insensitive must a parameter be to be fixed after fitting? What about a sensitivity test?
Parameters that were fixed were either gathered from previous studies or after observing little variance amongst the “best fit” values. The observed standard deviation for the best fit set was small compared to the mean of the distribution, and varying the parameter within two standard deviations of the mean did not meaningfully alter the output shown in Fig 2.
- L154, why 10 runs? This seems like a very small number especially for a relatively more complex model with more than a few processes.
Ten runs were selected because to do more so would have been computationally expensive. We noted that for the summary statistics listed in Fig 2, 10 runs with the same set of parameters typically yielded very similar values, indicating that this number of runs is sufficient.
- L154, is this what you mean “The same starting T cell distribution was used …”?
Thank you. This is corrected.
- L251, should “data” be “model estimated parameter”?
We agree and made this change.
- My opinion is that phrases like “extremely closely” (L258), “close representation” (L259), “extremely well” (L305) are not supported by the current results.
We have toned down this language, though we do feel as if the CCC listed for Figure 2 reveal that the model does reproduce key patterns in the data remarkable well for a simple model.
- L273-274 needs supporting reference.
References are already included for this sentence.
- L283, what is “outsized”?
We changed this to significant.
- L285-287, did you do any quantitative or qualitative test to check for similarity in patterns, such as correlation with temporal shift, dynamic time warping, etc.?
These results are already in our prior published paper which is cited.
Round 2
Reviewer 2 Report
Comments and Suggestions for Authors
The authors have addressed my comments satisfactorily.